# A Generic Class-agnostic Object Counting Network with Adaptive Offset Deformable Convolution

## Abstract

Class-agnostic object counting (CAC) aims at counting the number of objects in the unseen category in an image. In this paper, we design a generic class-agnostic object counting network with Adaptive Offset Deformable Convolution (AODC), which initially focus on the reference-less class-agnostic object counting task without any exemplar. Our method calculates the self-similarity maps of the image features and performing a 4D convolution on these maps, obtaining the adaptive offsets for the deformable convolution, so that the model can obtain complete information about the object at that location. Through this process, AODC is able to recognise objects of different scales in a same sample. In addition to this, we adopt our approach to both zero-shot setting and few-shot setting, the former with semantic text and the latter with visual exemplars as references. We conduct experiments on the few-shot object counting dataset FSC-147, as well as other large-scale datasets, and show that our method significantly outperforms state-of-the-art approaches on all the three settings.

## 1 Introduction

Object counting tasks have mainly focused on specific categories in the past, such as people (Shu et al., 2022; Abousamra et al., 2021; Cai et al., 2023), cars (Hsieh et al., 2017) or animals (Arteta et al., 2016; Zavrtanik et al., 2020). In contrast, class-agnostic object counting (CAC) has received considerable attention and development in recent years, especially after a dataset focusing on CAC is proposed (Ranjan et al., 2021). Not only does CAC require less data annotations than class-specific object counting, but it can also be applied to unseen categories. Using few provided visual references as exemplars in few-shot setting, CAC obtains a generalised counting model by learning the process of comparing the sample image and exemplars and regressing the feature representations.

In class-agnostic object counting, in addition to the few-shot setting, there is also a reference-less setting where no exemplars are used, and a zero-shot setting where the category names are used as references. Our approach focuses on all these domains, obtaining good performance for each setting. In the reference-less setting, the effectiveness of the method in (Ranjan & Nguyen, 2022) is highly dependent on the accuracy of the selected exemplars, which is difficult to guarantee. The performance of (Hobley & Prisacariu, 2022) relies on a pre-trained model that is trained with a large amount of data. For research related to zero-shot setting, previous work has proposed many effective approaches (Xu et al., 2023; Jiang et al., 2023; Kang et al., 2023; Amini-Naieni et al., 2023). However, as described in (Oquab et al., 2023; Paiss et al., 2023; Zhai et al., 2022; Amini-Naieni et al., 2023), general text encoding models, such as CLIP (Radford et al., 2021), lack the awareness of object spatial structure. This heavily limits the performance of existing zero-shot object counting methods. And in few-shot setting, many methods (Shi et al., 2022; Lin et al., 2022; Djukic et al., 2023) usually take the average length and width of the bounding boxes as a fixed scale when embedding object scale information, however, many objects within the sample images are of varying scales, and this processing is usually not effective enough to recognise objects that are extreme in scale.

Even if a person has never seen a certain category of objects, the human eyes can easily distinguish its general shape and area within the field of view. This is because an object typically has great

similarity in all parts of its body, which becomes more obvious when the object is in a background with a large difference in colour compared to itself. For class-agnostic object counting task where no exemplars are provided, the model cannot directly derive useful information from object features when faced with unseen categories. However, since feature similarity is unbiased towards any specific category, we can take advantage of the ability of human eyes to recognise objects by calculating and utilizing the self-similarity of the features to give the model the ability to recognise the shape and size of unseen category objects.

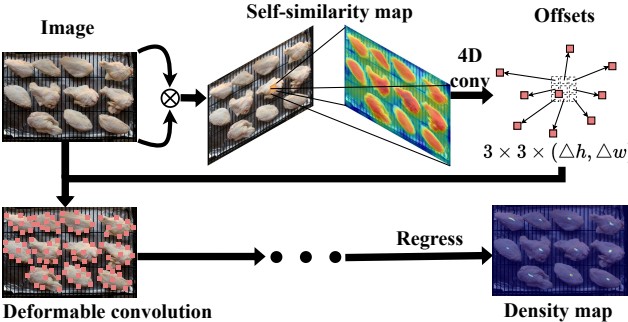

Figure 1: The general pipeline of our method. A set of $3 \times 3$ convolution kernels can be appropriately offset to cover the entire object.

In this paper, we present an end-to-end CAC model that can recognise the shapes and sizes of a category of objects, without the need for visual exemplars, additional training data or training stages. As illustrated in Fig. 1, after inputting an image and extracting features, the similarity value of each pixel in the feature map is computed with all other pixels to obtain a 4D similarity map. This is followed by a 4D convolution, which transforms the information about the self-similarity distribution of each point and its surroundings in the feature map into the horizontal and vertical offsets of that pixel point to the similarity boundaries. The adaptive offsets are used as the convolution kernel offsets to perform deformable convolution on the original feature map. This fuses the overall information of each object in the feature map into the centre position of the object, and further generates an accurate predicted density map in the final regression.

Having achieved recognition of the spatial structure of objects in the reference-less setting, then we can easily adapt our method to the zero-shot setting, where it is sufficient to embed the text into the same feature space as the image feature using a pre-trained semantic model, and before the computation of self-similarity maps we incorporate the semantic feature into the image feature map using several cross-attention modules to highlight features that are of the same category as the semantic feature. For the few-shot setting, since we have several more accurate visual exemplars, we can replace the self-similarity computation with a cross-similarity between the image feature map and the exemplar feature maps, which gives us more accurate offsets. The subsequent calculations for both settings are then the same as for the reference-less setting.

We conduct experiments on a large-scale few-shot object counting dataset FSC-147 for all the three settings, and the experimental results outperform recent state-of-the-art methods. In addition to this, we also perform cross-dataset validation on car counting dataset CARPK (Hsieh et al., 2017) and two subsets that have pre-trained object detectors on the COCO dataset (Lin et al., 2014), and the superior performance also demonstrates the generalized ability of our method.

Our contributions are summarized as follows: (1) We design a novel reference-less class-agnostic object counting network with Adaptive Offset Deformable Convolution (AODC), for counting unseen category objects without references. (2) We generalize AODC to zero-shot and few-shot settings to form a generic class-agnostic object counting network.

## 2 RELATED WORK

### 2.1 CLASS-SPECIFIC OBJECT COUNTING

Class-specific object counting focuses on counting a specific class of objects, such as crowd (Liang et al., 2023; Lin & Chan, 2023; Du et al., 2023), animals (Arteta et al., 2016), or cars (Hsieh et al., 2017), among which crowd counting has received the most extensive excavation. The earliest counting methods are based on object detection (Stewart et al., 2016; Wang & Wang, 2011), where the number of objects is obtained by counting the detection results. However, this kind of methods are less effective in identifying dense samples and require an additional object detection process.

To address this issue, counting methods based on density maps, which are called regression-based methods, are developed and widely adopted. These methods generate a density map and sum it to obtain the counting number. The model is trained by comparing the ground truth density map with the predicted density map.

Recent research in regression-based methods, such as (Cheng et al., 2022), utilizes locally connected multivariate Gaussian kernels as replacements for convolution filters. (Du et al., 2023) introduce domain-invariant and -specific crowd memory modules to extract disentangled domain-invariant/-specific features for each image. Moreover, a recent work (Liang et al., 2023) proposes knowledge transfer from a vision-language pre-trained model (CLIP) to unsupervised crowd counting tasks, eliminating the need for density map annotation.

## 2.2 CLASS-AGNOSTIC OBJECT COUNTING

Class-agnostic object counting (CAC) is first studied in (Lu et al., 2019) and has gained significant attention with the proposal of a challenged dataset (Ranjan et al., 2021). In contrast to class-specific object counting task, CAC can exhibit great generality by using a few exemplars to count objects of unseen categories.

Several terrific methods have been proposed for CAC. As the first novel in the study of CAC, GMN (Lu et al., 2019) integrates support and query features, subsequently applying regression to forecast a density map from this amalgamation. Over the next few years, many methods in this area were proposed (Ranjan et al., 2021; Shi et al., 2022; Liu et al., 2022; Djukic et al., 2023; Wang et al., 2024), leading to significant development of CAC tasks. The most recent approach, CACViT (Wang et al., 2024) proposes a ViT-based extract-and-match paradigm for CAC, and introduces aspect-ratio-aware scale embedding and magnitude embedding to compensate for the information loss.

For the setting without references, RepRPN-C (Ranjan & Nguyen, 2022) proposes a two-stage counter. It consists of a novel region proposal network for finding exemplars from repetitive object classes and a density estimation network to estimate the density map corresponding to each exemplar. RCC (Hobley & Prisacariu, 2022) is based on the confirmed intuition that well-trained vision transformer features are both general enough and contextually aware enough to implicitly understand the underlying basis of counting. It is important to note that both CounTR (Liu et al., 2022) and LOCA are tested in a setting without references and achieve positive performance.

The zero-shot object counting is first investigated by (Xu et al., 2023), which proposes a reference exemplar using the existing few-shot object counting methods and uses them for further counting. following this work, several brand new zero-shot object counting methods have been proposed (Jiang et al., 2023; Kang et al., 2023; Amini-Naieni et al., 2023) and have performed well.

## 2.3 GENERALIZED LOSS

In order to improve the performance of crowd counting methods, (Wan et al., 2021) proposes a generalised loss function based on unbalanced optimal transport. In class-agnostic object counting, (Lin et al., 2022) adopts this loss function and proposes a scale-sensitive generalised loss function that can be applied to different loss calculation methods depending on the object categories of different scales.

# 3 METHODOLOGY

## 3.1 PRELIMINARIES

Class-agnostic object counting divides the classes of the dataset into base classes $C_{base}$, which has been seen during training, and the unseen classes $C_{novel}$, where $C_{base}$ and $C_{novel}$ do not intersect, and the task of CAC is to be able to count samples containing objects from these unseen classes $C_{novel}$ at test time even if they have not been seen during training. For the few-shot setting, the model has to count the number of objects of the same category in the image based on the given $K$ visual exemplars, whereas in the zero-shot setting, the exemplars are textual information about this category, and the model needs to embed the category text into the feature space and compare

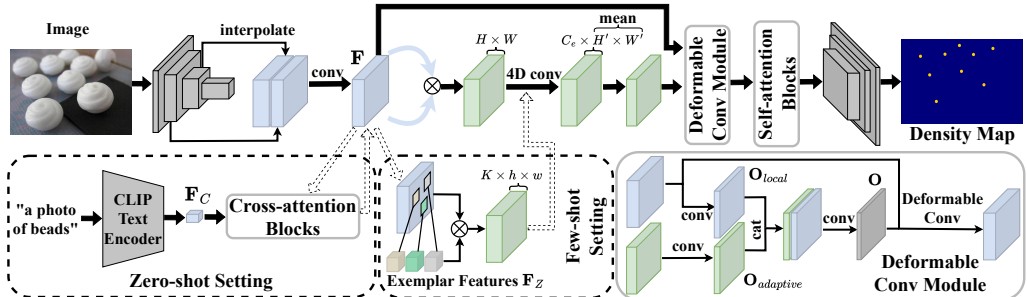

Figure 2: The whole architecture of the proposed AODC framework.

with the image features. Additionally, when considering the reference-less setting, neither visual nor textual exemplars are provided, and the model needs to identify the possible objects in the image.

**Centre-pivot 4D Convolution.** 4D convolution has been proposed and studied in previous work (Rocco et al., 2018; Yang & Ramanan, 2019; Min et al., 2021). In this paper, we use centre-pivot 4D convolution from (Min et al., 2021) to speed up the convolution process. For the position $(\mathbf{x}, \mathbf{x}')$ of a point in a 4D map $\mathbf{M}$, define two sets $\mathbf{P}(\mathbf{x})$ and $\mathbf{P}(\mathbf{x}')$, which contain the points in a neighbourhood of the size of the convolution kernel around $\mathbf{x}$ and $\mathbf{x}'$, and the centre-pivot 4D convolution can be formulated as:

$$(\mathbf{M} * \mathbf{k})(\mathbf{x}, \mathbf{x}') = \sum_{\mathbf{p}' \in \mathbf{P}(\mathbf{x}')} \mathbf{M}(\mathbf{x}, \mathbf{p}') \mathbf{k}^0(\mathbf{p}' - \mathbf{x}') + \sum_{\mathbf{p} \in \mathbf{P}(\mathbf{x})} \mathbf{M}(\mathbf{p}, \mathbf{x}') \mathbf{k}^1(\mathbf{p} - \mathbf{x}), \tag{1}$$

where $\mathbf{k}^0$ and $\mathbf{k}^1$ are the 2D kernels on 2D slices of 4D tensor $\mathbf{M}(\mathbf{x}, :)$ and $\mathbf{M}(:, \mathbf{x}')$, and $\mathbf{k} = [\mathbf{k}^0, \mathbf{k}^1]$.

### 3.2 Basic Reference-less Framework

As shown in Fig. 2, Our framework is originally designed to solve the reference-less object counting task, and both the zero-shot setting and the few-shot setting are adaptation of this foundational framework. Given a sample image $\mathbf{X}$, we need to identify and count the objects of the corresponding category that may be present in the image without any reference, obtaining the predicted density map $\mathbf{D}_p$, and the counting number $C_p$ obtained by summing $\mathbf{D}_p$.

**Feature Extraction.** we choose the pre-trained ResNet-50 (He et al., 2016) as the AODC backbone. Inspired by (Djukic et al., 2023), we use the similar processing to extract the output feature maps of the second and third layers, all resize to the size $H \times W$ of the second layer feature map, and concatenate them together before a layer of convolution to channel $C$, and obtain the extracted feature map $\mathbf{F} \in \mathbb{R}^{C \times H \times W}$.

**Adaptive Offsets.** In order to perform deformable convolution with different offsets depending on the object scale, we need to compute the corresponding convolution offsets based on the similarity distribution between each pixel occupied by the object and its surroundings. First let the feature at each position in $\mathbf{F}$ be multiplied with all the features:

$$\mathbf{S}(\mathbf{x}^0, \mathbf{x}^1) = \text{ReLU}\left(\frac{\mathbf{F}(\mathbf{x}^0) \cdot \mathbf{F}(\mathbf{x}^1)}{\|\mathbf{F}(\mathbf{x}^0)\| \|\mathbf{F}(\mathbf{x}^1)\|}\right). \tag{2}$$

Here, $\mathbf{x}^0$ and $\mathbf{x}^1$ denote 2-dimensional spatial positions of the two feature maps. '·' denotes vector dot product. Then we obtain the self-similarity map $\mathbf{S} \in \mathbb{R}^{H \times W \times H \times W}$.

To transform this 4D self-similarity map information into adaptive offsets at each position, we apply 4D convolution on it. After convolution, $\mathbf{S}$ is transformed into an adaptive offset feature map $\mathbf{F}_{offset}$:

$$\mathbf{F}_{offset} = f_e(\mathbf{S}) \in \mathbb{R}^{C_e \times H \times W \times H' \times W'}, \tag{3}$$

where $C_e$ is the offset feature channel length. $f_e(\cdot)$ is an encoding module formed by concatenating several layer combinations. Each combination consists of a 4D convolutional layer, group normalization (Wu & He, 2018), and ReLU activation function.

Since the similarity of the two features is a single value, the input length to $f_e(\cdot)$ is 1 and is convolved to get the offset feature of length $C_e$. Multiple large strides in the convolution module reduce the last two dimensions of the map to $H' \times W'$, which we then take the mean of to get $\mathbf{F}_{offset} \in \mathbb{R}^{C_e \times H \times W}$.

**Deformable Convolution Module.** Inspired by (Lin et al., 2022), we design a similar deformable convolution module to convolve $\mathbf{F}$ according to the offset information of $\mathbf{F}_{offset}$. At first, $\mathbf{F}$ and $\mathbf{F}_{offset}$ are each passed through a convolution module to obtain a local offset map $\mathbf{O}_{local}$ and an adaptive offset map $\mathbf{O}_{adaptive}$, respectively, and then the two maps are concatenated together and passed through a fusion convolution module to obtain the final offset map $\mathbf{O} \in \mathbb{R}^{C_{offset} \times H \times W}$, where $\mathbf{C}_{offset} = k \times k \times (\triangle h, \triangle w)$ Indicates the offsets of the convolution kernel of size $k \times k$ at each position in the feature map.

We define a convolution kernel $\mathbf{W}$ with size $k \times k$, and then perform deformable convolution on $\mathbf{F}$ with $\mathbf{O}$. For the point set $\mathbf{P_q}$ in a neighbourhood of this kernel size near position $\mathbf{q}$, the output $\mathbf{F}_D(\mathbf{q})$ is calculated as:

$$\mathbf{F}_D(\mathbf{q}) = \sum_{\mathbf{q}' \in \mathbf{P_q}} \mathbf{F}(\mathbf{q}' + \mathbf{O}(\mathbf{q}, \mathbf{q}' - \mathbf{q})) \mathbf{W}(\mathbf{q}' - \mathbf{q}). \tag{4}$$

The obtained feature map $\mathbf{F}_D$ is further fed into a module formed by concatenating several self-attention blocks.

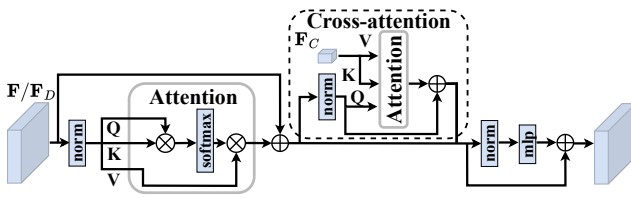

Figure 3: The architecture of the two attention blocks. The cross-attention block contains all the structures in the figure, while the self-attention block excludes the cross-attention part.

**Self-attention Block.** After deformable convolution, some of the locations of the feature map contain information about the objects, and in order to highlight and standardise the features at these locations. We input $\mathbf{F}_D$ into a sequence of several self-attention blocks, where features with objects of the same category are clustered together according to their degree of similarity. The structure of a single self-attention block is shown in Fig. 3, where the feature map $\mathbf{F}_D$ is simultaneously fed into the attention mechanism as Query, Key, and Value. The module output is then fed into the next self-attention block and the final output goes into the regression head to compute the predicted density map.

**Regression.** The regression head consists of several combinations of layers stacked on top of each other, with each combination consisting of a $3 \times 3$ convolutional layer, a ReLU activation layer, and an upsampling layer. The upsampling doubles the feature map size, and several upsampling layers scale the feature map to the size of the original image. The final tail is a $1 \times 1$ convolutional layer and ReLU activation layer, which regresses the feature channel to a density value and outputs the predicted density map $\mathbf{D}_p$.

### 3.3 ZERO-SHOT SETTING

Objects from other categories are often counted together in reference-less setting because category information is not provided. The text provided in zero-shot setting can be used to highlight the category objects that need to be counted and avoid interference from other categories. In order to embed the category name $C$ in textual form into the feature space, we use a pre-trained CLIP (Radford et al., 2021) model to transform $C$ into a feature vector $\mathbf{F}_C$. $\mathbf{F}_C$ is fed into several cross-attention blocks together with $\mathbf{F}$ to complete the fusion of feature information. The structure of the cross-attention block is shown in Fig.3, and the difference with the structure of the self-attention block is that the cross-attention block has one more cross-attention computation process in the middle part. $\mathbf{F}$ is used as Query and $\mathbf{F}_C$ is used as Key and Value in the cross-attention.

### 3.4 FEW-SHOT SETTING

The bounding boxes of the location of the $K$ exemplars are provided in the few-shot setting, from which we can extract the exemplar features $\mathbf{F}_Z \in \mathbb{R}^{K \times C \times h \times w}$ from the image feature map $\mathbf{F}$ using

the ROIAlign method (He et al., 2017). Because $\mathbf{F}_Z$ contain more accurate object information, we replace the self-similarity with the cross-similarity between $\mathbf{F}$ and $\mathbf{F}_Z$, and replace the second feature map in Eq.2 with $\mathbf{F}_Z$ to compute to obtain the cross-similarity map $\mathbf{S}'$:

$$\mathbf{S}'\left(\mathbf{x}^0, \mathbf{x}'\right) = \text{ReLU}\left(\frac{\mathbf{F}\left(\mathbf{x}^0\right) \cdot \mathbf{F}_Z\left(\mathbf{x}'\right)}{\left\|\mathbf{F}\left(\mathbf{x}^0\right)\right\| \left\|\mathbf{F}_Z\left(\mathbf{x}'\right)\right\|}\right). \tag{5}$$

$\mathbf{S}'$ instead of $\mathbf{S}$ is input to the subsequent 4D convolution module for computation, followed by the same workflow in reference-less setting. It is worth noting that since we have $K$ exemplars, the channel length of input to the 4D convolution module changes from 1 to $K$ in this setting.

### 3.5 GENERALIZED LOSS

In order to speed up the convergence of crowd counting and improve the performance, (Wan et al., 2021) proposes a generalized loss to measure the distance between the predicted density map and the ground truth dot labels, which we also employ in this paper for supervised training of our model.

We define the predicted density map and the ground truth dot labels as $\mathbf{A} = \{(a_i, \mathbf{x}_i)\}_{i=1}^{n}$ and $\mathbf{B} = \{(b_j, \mathbf{y}_j)\}_{j=1}^{m}$, respectively, where $a_i$ is the predicted density value at location $\mathbf{x}_i \in \mathbb{R}^2$ and $n$ is the number of pixels, and we use this to set the predicted density map to be $\mathbf{a} = [a_i]_i$. $\mathbf{y}_j$ and $b_j$ are the location of the dot labels and the number of objects at that location, which we simplify to $\mathbf{b} = [b_j]_j = 1_m$. The loss function is formulated as:

$$L\left(\mathbf{A}, \mathbf{B}\right) = \min_{\mathbf{D}} \langle \mathbf{C}, \mathbf{D} \rangle - \varepsilon H\left(\mathbf{D}\right) + \tau \left\|\mathbf{D}1_m - \mathbf{a}\right\|_2^2 + \tau \left\|\mathbf{D}^T 1_n - \mathbf{b}\right\|_1. \tag{6}$$

$\mathbf{C}$ denotes the cost required to move the predicted density to the ground truth dot label, $\mathbf{D}$ is the transport matrix for cost calculation, and $H\left(\mathbf{D}\right) = -\sum_{ij} D_{ij} \log D_{ij}$ is the entropic regularization. $\varepsilon$ and $\tau$ are two hyper-parameters to be tuned.

## 4 EXPERIMENTS

### 4.1 DATASETS AND METRICS

**Datasets. FSC-147** is a multi-class few-shot object counting dataset that is comprehensive in nature. It comprises 6,135 images that cover 89 different object categories. The dataset is further divided into training, validation, and testing subsets, each containing 29 non-overlapping object categories. the number of objects in images ranges from a minimum of 7 objects to a maximum of 3,731 objects, with an average of 56 objects per image. Additionally, each image in the dataset is accompanied by three to four exemplar images, all marked with bounding boxes for easier identification.

**Metrics.** The metrics used to evaluate our AODC method are Mean Average Error (MAE) and Root Mean Squared Error (RMSE), which are commonly used in object counting tasks, and their formulas are defined as follows:

$$MAE = \frac{1}{N} \sum_{i=1}^{N} \left|C_{pred}^i - C^i\right|, RMSE = \sqrt{\frac{1}{N} \sum_{i=1}^{N} (C_{pred}^i - C^i)^2}, \tag{7}$$

where $N$ is the number of all the sample images, $C^i$ and $C_{pred}^i$ are the ground truth and the predicted number of objects for $i$-th image.

### 4.2 IMPLEMENTATION DETAILS

**Architecture Details.** The images in our method are uniformly resized to $384 \times 576$ and then fed into the ResNet-50 pre-trained model, the feature channel length of the output feature map is 512. 4D convolution module has 4 layers and the output feature length is 8, 16, 32, 64, thus the offset feature channel length $C_e$ is 64. The number of self-attention blocks is 3, and the number of cross-attention blocks in the zero-shot setting is 2. The exemplar feature size in the few-shot setting is $32 \times 32$.

Table 1: Comparison with state-of-the-art approaches on the FSC-147 dataset. '†' means that the method uses a customized text description.

| Scheme | Methods | Backbone | exemplars | Val | | Test | |
|---|---|---|---|---|---|---|---|
| | | | | MAE | RMSE | MAE | RMSE |
| Few-shot | FamNet (Ranjan et al., 2021) | ResNet-50 | Visual Exemplars | 23.75 | 69.07 | 22.08 | 99.54 |
| | BMNet+ (Shi et al., 2022) | ResNet-50 | Visual Exemplars | 15.74 | 58.53 | 14.62 | 91.83 |
| | SAFECount (You et al., 2023) | ResNet-18 | Visual Exemplars | 15.28 | 47.20 | 14.32 | 85.54 |
| | SPDCN (Lin et al., 2022) | VGG-19 | Visual Exemplars | 14.59 | 49.97 | 13.51 | 96.80 |
| | CounTR (Liu et al., 2022) | ViT/ConvNet | Visual Exemplars | 13.13 | 49.83 | 11.95 | 91.23 |
| | LOCA (Djukic et al., 2023) | ResNet-50 | Visual Exemplars | 10.24 | 32.56 | 10.79 | 56.97 |
| | CACViT (Wang et al., 2024) | ViT | Visual Exemplars | 10.63 | 37.95 | **9.13** | **48.96** |
| | AODC(Ours) | ResNet-50 | Visual Exemplars | **10.09** | **30.88** | 10.64 | 65.17 |
| Zero-shot | ZSC (Xu et al., 2023) | ResNet-50/CLIP | Text | 26.93 | 88.63 | 22.09 | 115.17 |
| | CLIP-Count (Jiang et al., 2023) | ViT/CLIP | Text | 18.79 | 61.18 | 17.78 | 106.62 |
| | VLCounter (Kang et al., 2023) | ViT/CLIP | Text | 18.06 | 65.13 | 17.05 | 106.16 |
| | CounTX (Amini-Naieni et al., 2023) | ViT/CLIP | Text | 17.70 | 63.61 | 15.73 | 106.88 |
| | CounTX† (Amini-Naieni et al., 2023) | ViT/CLIP | Text | 17.10 | 65.61 | 15.88 | 106.29 |
| | AODC(Ours) | ResNet-50/CLIP | Text | **14.27** | **47.12** | **14.72** | **104.90** |
| Reference-less | RepRPN-C (Ranjan & Nguyen, 2022) | ResNet-50 | None | 29.24 | 98.11 | 26.66 | 129.11 |
| | RCC (Hobley & Prisacariu, 2022) | ViT | None | 17.49 | 58.81 | 17.12 | 104.53 |
| | CounTR (Liu et al., 2022) | ViT/ConvNet | None | 17.40 | 70.33 | **14.12** | 108.01 |
| | LOCA (Djukic et al., 2023) | ResNet-50 | None | 17.43 | 54.96 | 16.22 | 103.96 |
| | AODC (Ours) | ResNet-50 | None | **14.54** | **48.68** | 14.84 | **103.67** |

**Training Details.** Our model is trained end-to-end and the backbone parameters are frozen. We apply AdamW (Loshchilov & Hutter, 2017) as the optimizer with a learning rate of $1 \times 10^{-4}$ and the learning rate decays with a rate of 0.95 after each epoch. The parameters $\varepsilon$ and $\tau$ in the generalized loss function are set to 5 and 0.01, respectively. The batch size is 4 and the model is trained on a single RTX A6000 for 100 epochs, which cost about 15 hours.

### 4.3 COMPARISON WITH STATE OF THE ART

We perform evaluation experiments of AODC on the few-shot object counting dataset FSC-147, where we conduct experiments on all three settings and compare them to the state-of-the-art methods on each setting separately and summarize the results in Tab. 1.

In the few-shot setting, we mainly compare AODC with the most recent method CACViT (Wang et al., 2024), which is based on ViT pre-trained model on MAE and uses several post-hoc error compensation routines as applied in CounTR (Liu et al., 2022) to reduce the error during model testing. Even so, AODC still manages to outperform CACViT in the validation set, obtaining a 5.5% improvement on MAE and 18.6% on RMSE.

In the zero-shot setting, the existing methods have a bottleneck in performance that is difficult to break through due to the lack of recognition of the spatial structure of the objects. AODC has a significant improvement in performance relative to these methods because of the acquisition of spatial structure information based on the basic reference-less framework. Compared to the first zero-shot object counting method ZSC (Xu et al., 2023), AODC has a huge improvement of 47.0% on MAE and 46.8% on RMSE on the validation set, as well as 33.4% on MAE and 9.0% on RMSE on the test set. Compared to the recent state-of-the-art method CounTX (Amini-Naieni et al., 2023), AODC achieves 19.4% on MAE and 25.9% on RMSE on the validation set. On the test set, AODC also shows some improvement. Even compared to CounTX that uses special text descriptions, AODC still outperforms it on all metrics.

In the reference-less setting, we compare and analyze each metric separately as several state-of-the-art methods (Hobley & Prisacariu, 2022; Liu et al., 2022; Djukic et al., 2023) have similar performance. In the validation set, the MAE metrics of all three state-of-the-art methods are around 17.4, AODC exhibits an 16.4% improvement. And for RMSE, AODC achieves a 11.4% to 30.8% improvement. In the test set, on MAE, except for CounTR, compared to the other two methods AODC shows 8.5% to 13.3% improvement. While for RMSE, AODC exceeds all state-of-the-art methods.

**Qualitative Results.** In each setting, we visualize some of the prediction results of AODC and a state-of-the-art method for comparison and show them in Fig. 4. It can be seen that AODC has good prediction ability in all settings for both dense and sparse samples. Since AODC has the ability to adaptively recognize objects without relying on a reference, it has a good abil-

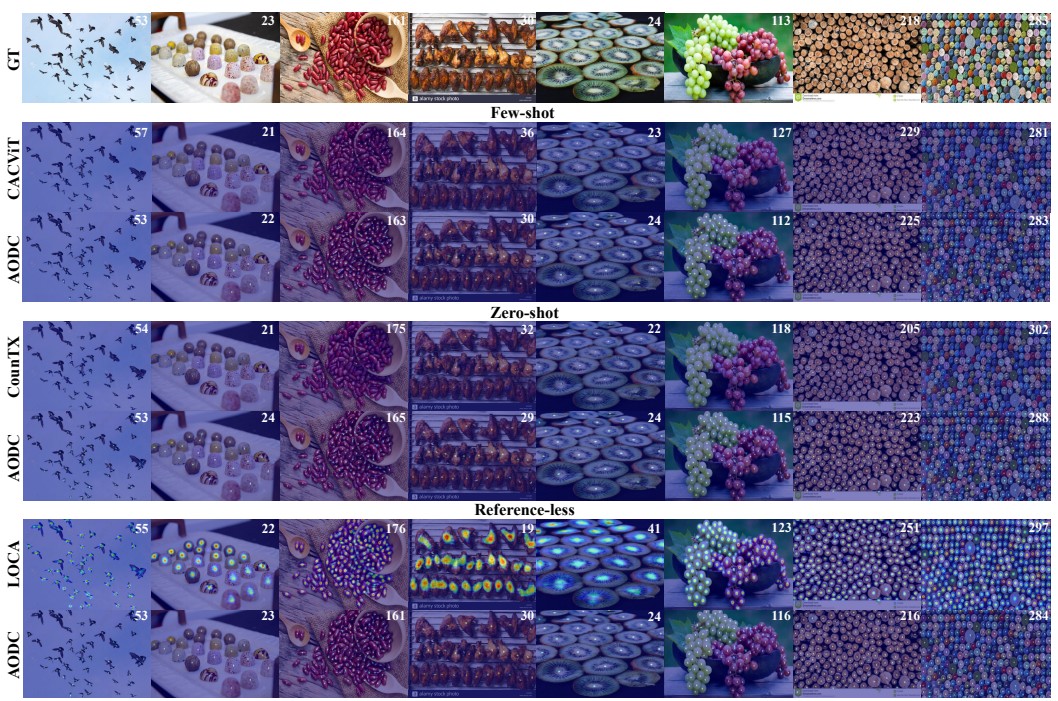

Figure 4: Qualitative results on the FSC-147 dataset.

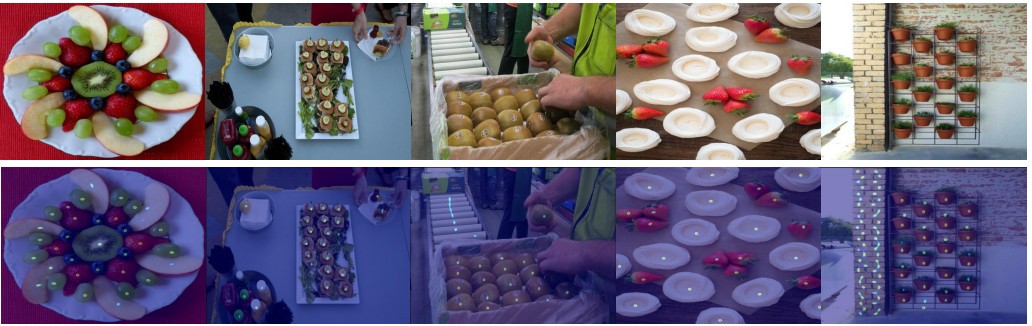

Figure 5: Results for samples containing objects from multiple categories in the reference-less setting. Due to the lack of category information, the model recognizes as many objects present in the image as possible.

ity to recognize mutilated or occluded objects. For example, CounTX in columns 5 and 7 does not recognize some objects that are only partially revealed, resulting in a final prediction count that is too small. LOCA, on the other hand, focuses too much on counting dense samples, which makes its prediction ability for samples with larger objects and sparse distribution poor.

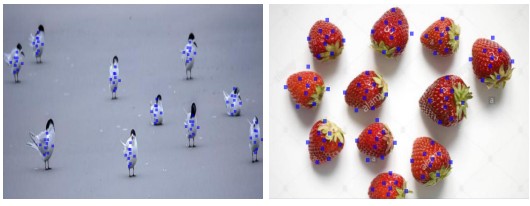

Figure 6: Visualization of deformable convolution kernel positions for each object location.

For example, in the 4 and 5 columns, the prediction results of LOCA show a large error. Comparatively AODC has a good ability to recognize both large or small objects and predicts the correct densities for objects of different scales.

A issue that exists in the reference-less setting is that, due to the lack of reference, the model is unable to recognize which categories of objects in the image should be counted and which should not, and will count all objects as far as possible, as in Fig. 5. On the other hand, without providing category information, all categories are supposed to be counted. However, since the

Table 2: Comparison with the state-of-the-art approaches on the cross-datasets.

| Scheme | Methods | CARPK | | Val-COCO | | Test-COCO | |
|---|---|---|---|---|---|---|---|
| | | MAE | RMSE | MAE | RMSE | MAE | RMSE |
| Few-shot | FamNet (Ranjan et al., 2021) | 28.84 | 44.47 | 39.82 | 108.13 | 22.76 | 45.92 |
| | BMNet (Shi et al., 2022) | 10.44 | 13.77 | 26.55 | 93.63 | 12.38 | 24.76 |
| | LOCA (Djukic et al., 2023) | 9.97 | 12.51 | **16.86** | 53.22 | 10.73 | 31.31 |
| | CACViT (Wang et al., 2024) | 8.30 | 11.18 | 20.00 | 58.97 | **8.55** | **18.42** |
| | AODC (Ours) | **7.08** | **9.68** | 18.24 | **52.67** | 11.13 | 26.05 |
| Zero-shot | CLIP-Count (Jiang et al., 2023) | 11.96 | 16.61 | 26.43 | 85.13 | 16.35 | 38.86 |
| | CounTX (Amini-Naieni et al., 2023) | 11.64 | 14.85 | 29.39 | 101.56 | 12.15 | **25.49** |
| | AODC (Ours) | **7.29** | **10.15** | **22.58** | **67.49** | **11.35** | 28.61 |
| Reference-less | RCC (Hobley & Prisacariu, 2022) | 12.31 | 15.40 | 23.44 | 68.21 | 13.07 | 28.01 |
| | AODC (Ours) | **7.31** | **10.34** | **22.74** | **65.27** | **12.32** | **27.16** |

FSC-147 dataset only labels one category for each sample, it is not possible to accurately measure the ability to count multiple categories.

To show more intuitively the effect of AODC in capturing and recognizing objects without references, we display the offset convolution kernel positions obtained by the model for each ground truth point location on the original image. As shown in Fig. 6, for objects of different scales and shapes, our method is able to offset the convolution kernels to the appropriate positions to enclose the whole objects as much as possible, thus obtaining complete and accurate information.

### 4.4 CROSS-DATASET GENERALIZATION

In addition to the few-shot object counting dataset FSC-147, we also cross-evaluate the performance of AODC on the car-counting dataset CARPK (He et al., 2017) and the COCO (Lin et al., 2014) subsets of FSC-147. CARPK contains 1448 images from several parking lots taken from a bird's view, and the training set contains data from three parking lot scenes, while the test set has data from one other scene. We put the AODC model trained on FSC-147 onto the test set of CARPK for evaluation and the car category samples in the training set are eliminated. FSC-147 provides the subsets that have pre-trained object detectors on the COCO dataset, Val-COCO and Test-COCO, for comparing the performance on them with the object detection methods. The two subsets contain 277 and 282 images, respectively, and we validate the performance of AODC on these two sets. All the results are shown in Tab. 2.

Overall, AODC on the CARPK dataset outperforms all state-of-the-art methods by a large margin, and it is noteworthy that the experimental results even on reference-less setting outperform the most recent state-of-the-art CACViT on the few-shot setting, which demonstrates the strong generalization ability of AODC. For Val-COCO and Test-COCO, AODC also has superior performance. The error values for most of the metrics of AODC are lower than the existing state-of-the-art methods.

### 4.5 ABLATION STUDY

To determine the contribution of the adaptive offset convolution to the model, we remove the adaptive offsets in the deformable convolution module to perform ablation experiments. In addition to this, AODC also has a self-attention module and generalized loss to further enhance the model in addition to the necessary frame components. In order to verify the specific enhancement effect of these components, we conduct the corresponding ablation experiments and display the results in Tab. 3. With or without a single component each forms four sets of comparison experiments: contrast the addition of this component to a baseline model without any components, and the addition of this component to a model with only one of the other two components alone, with the addition of this component to a model with the other two components to form a complete AODC model. The results of these experiments reflect the effect of each component on the overall enhancement of the model.

**Adaptive Offsets.** After removing the adaptive offsets, the deformable convolution can only obtain the scale information of the object through the feature itself, which limits the recognition performance of the model to a great extent. The experimental results show that the performance of the model with adaptive offsets is significantly enhanced. An improvement of up to 21.2% on MAE

and 27.8% on RMSE is obtained on the validation set, as well as an improvement of up to 18.1% on MAE and 6.1% on RMSE on the test set.

Table 3: Ablation studies on the FSC-147 dataset. 'G-Loss' means Generalized Loss, 'Self-attn' means self-attention mechanism and 'Adapt-O' means Adaptive Offsets.

| Adapt-O | Self-attn | G-Loss | Val | | Test | |
|---|---|---|---|---|---|---|
| | | | MAE | RMSE | MAE | RMSE |
| ✗ | ✗ | ✗ | 25.29 | 70.45 | 24.68 | 115.92 |
| ✓ | ✗ | ✗ | 22.33 | 70.46 | 21.85 | 112.94 |
| ✗ | ✓ | ✗ | 23.44 | 68.19 | 23.56 | 114.31 |
| ✗ | ✗ | ✓ | 20.87 | 68.52 | 20.33 | 109.76 |
| ✓ | ✗ | ✓ | 17.10 | 63.74 | 17.69 | 107.43 |
| ✗ | ✓ | ✓ | 18.45 | 67.40 | 18.13 | 110.35 |
| ✓ | ✓ | ✗ | 20.71 | 65.43 | 19.49 | 108.02 |
| ✓ | ✓ | ✓ | 14.54 | 48.68 | 14.84 | 103.67 |

**Self-attention Module.** The clustering effect of the self-attention mechanism on features of the same category of objects is significant, with the model that employs self-attention obtaining up to 15.0% on MAE and 23.6% on RMSE in the validation set, and up to 16.1% on MAE and 6.0% on RMSE in the test set, as opposed to directly regressing on the deformable convolved features. This operation not only standardizes the information representation of the individual objects, but also provides a good separation between the objects and the background features.

**Generalized Loss.** Since the principle of AODC is to localize to the center of the object by deformable convolution with the same offset as the object size, the generalized loss based on point labels is more suitable for our method than the MSE loss. It can be observed from the experimental results that the use of generalized loss gives a very significant improvement to the model, obtaining up to 29.8% on MAE and 25.6% on RMSE on the validation set, and up to 23.9% on MAE and 5.3% on RMSE on the test set.

Table 4: Ablation studies of the number of attention blocks in the two attention modules.

| | Blocks | 1 | 2 | 3 | 4 | 5 | 6 |
|---|---|---|---|---|---|---|---|
| Self-attn | MAE | 16.20 | 15.83 | **14.54** | 15.23 | 15.51 | 15.72 |
| | RMSE | 60.45 | 58.28 | **48.68** | 51.94 | 53.42 | 52.83 |
| Cross-attn | MAE | 14.89 | **14.27** | 14.43 | 15.15 | 15.87 | 16.19 |
| | RMSE | 59.04 | **47.12** | 52.48 | 59.80 | 61.36 | 65.24 |

Both the self-attention module and the cross-attention module each contain a sequence of several attention blocks, and we conduct the corresponding experiments on the validation set of FSC-147 to verify the optimal number of blocks for each. As shown in Tab. 4, as the number of self-attention blocks increases from 1 to 6, the model performance gradually becomes stronger and reaches an optimum at 3, after which the metrics begin to gradually increase, indicating that the model begins to overfit. The case of cross-attention is similar, after the optimal performance is reached at a block number of 2, the model performance is not improved as the number of blocks increases and the model complexity rises, so the optimal number of cross-attention blocks can be determined from this analysis.

## 5 CONCLUSION

We present a novel generic network for class-agnostic object counting task with adaptive offset deformable convolution (AODC), which is initially designed for solving the counting task on the reference-less setting and can be further generalized to the zero-shot and few-shot settings. AODC obtains the scale offsets of the object corresponding to each position by using 4D convolution on the self-similarity maps of the image features, and using the obtained offsets to perform deformable convolution on the image features to capture the entire object, which in turn is regressed to obtain an accurate predicted density map. Experiments are conducted on multiple datasets and the results demonstrate that we achieve state-of-the-art performance on all the three settings.

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

# A APPENDIX

## A.1 COMPUTATIONAL COST

Table 5: Comparison of computational cost.

| Scheme | Methods | GFLOPs | Params(M) | Epochs |
|---|---|---|---|---|
| Few-shot | BMNet (Shi et al., 2022) | 239.92 | 13.08 | 300 |
| | CounTR (Liu et al., 2022) | 84.75 | 99.26 | 1000 |
| | LOCA (Djukic et al., 2023) | 395.95 | 32.46 | 200 |
| | CACViT (Wang et al., 2024) | 88.80 | 99.24 | 200 |
| | AODC (Ours) | 109.12 | 32.40 | 100 |
| Zero-shot | CLIP-Count (Jiang et al., 2023) | 246.00 | 236.02 | 200 |
| | VLCounter (Kang et al., 2023) | 63.98 | 88.53 | 200 |
| | CounTX (Amini-Naieni et al., 2023) | 43.88 | 93.82 | 1000 |
| | AODC (Ours) | 139.12 | 81.23 | 100 |
| Reference-less | RCC (Hobley & Prisacariu, 2022) | 16.76 | 21.67 | 80 |
| | LOCA (Djukic et al., 2023) | 49.49 | 31.85 | 200 |
| | AODC (Ours) | 111.78 | 47.36 | 100 |

In order to show the complexity and computational cost of our method, we record the values of FLOPs and the number of parameters for AODC and some other state-of-the-art methods and display them in Tab. 5.

From the values in the table, it can be seen that in the few-shot setting, the computational cost of our method is smaller than that of LOCA and BMNet, and the number of parameters is almost the same with LOCA. The number of epochs we need for training is half of that of other methods, which makes the training of AODC more efficient and faster. In the zero-shot setting, the number of parameters is not much different between the methods, while the training speed of AODC is better than CLIP-Count and CounTX (CounTX is not as efficient because the number of epochs needed for finetune is too large). The computational cost of CounTR on the Reference-less setting are not listed here because they are almost the same with which on few-shot setting. The reason why the computational cost of RCC is so small is because its performance mainly relies on a large amount of data trained pre-training model, RCC does not need additional structure and computation, direct regression can obtain certain results. Our training speed is similar compared to LOCA, while the number of parameters is slightly higher.

Table 6: Analysis of layer combinations of extracted features.

| Feature Layers | Val | | Test | |
|:---:|:---:|:---:|:---:|:---:|
| | MAE | RMSE | MAE | RMSE |
| 2 | 17.45 | 62.84 | 17.29 | 109.21 |
| 2+3 | **14.54** | **48.68** | **14.84** | **103.67** |
| 2+4 | 18.47 | 64.92 | 17.86 | 108.52 |
| 2+3+4 | 15.42 | 52.95 | 15.80 | 104.31 |

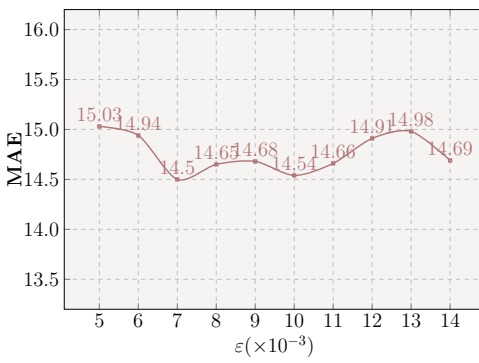

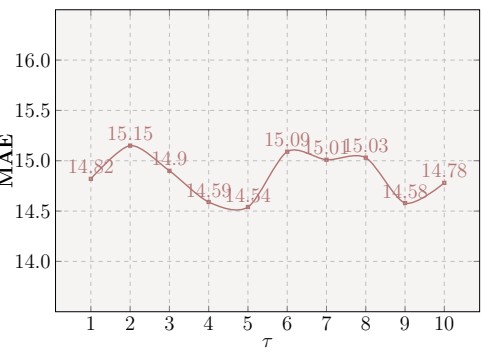

(a) Ablation of the hyper-parameter $\varepsilon$

(b) Ablation of the hyper-parameter $\tau$

Figure 7: Ablation study of hyper-parameters in generalized loss.

In overall, the computational cost and training efficiency of AODC is moderate and perfectly acceptable for the performance it achieves.

## A.2 ADDITIONAL ABLATION STUDY

For the multi-layer features from the pre-trained ResNet-50 backbone, we select different combinations of these feature layers for the experiments. As shown in Tab. 6, we divide the feature layers into four combinations and display the experimental results in the table. Among them, the second layer of features is necessary because our method is carried out on the spatial size of this layer. The experimental results show that combining the features of layers 2 and 3 gives the best performance, while adding the fourth layer leads to a decrease in performance. This may be due to the fact that too much spatial information is already lost in layer 4 and the oversized dimensions instead contain redundant information that is not needed for the counting task, making the model performance negatively affected. Dropping the layer 4 features not only optimizes performance, but also keeps computational complexity at a relatively acceptable level.

Two hyperparameters $\varepsilon$ and $\tau$ are defined in our quoted generalized loss function, and in order to determine the impact of these two parameters on the performance of our model, we conduct the corresponding experiments on the FSC-147 validation set and show the results in Fig. 7. The performance of the model does not fluctuate too much with parameter variations for both $\varepsilon$ and $\tau$, and most of the value variations are within the range of 0.5. This indicates that the training effect of the model is insensitive to parameter changes in the case that $\varepsilon$ and $\tau$ do not take particularly extreme values, ensuring the robustness of our method.

## A.3 MORE EXPERIMENTAL RESULTS

We visualize more visualization of deformable convolutional kernel positions, more qualitative results in FSC-147 and some qualitative results in CARPK and display them in Fig. 8, Fig. 9 and Fig. 10 for the readers to refer to.

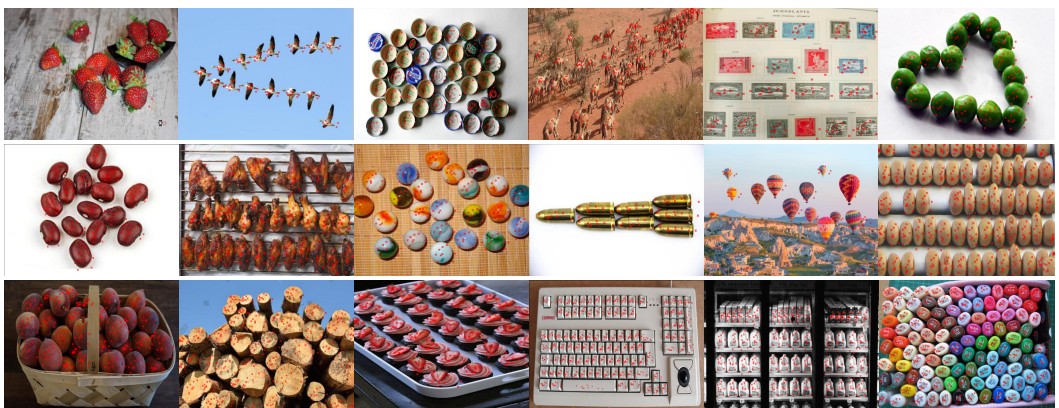

Figure 8: More visualization of deformable convolutional kernel positions.

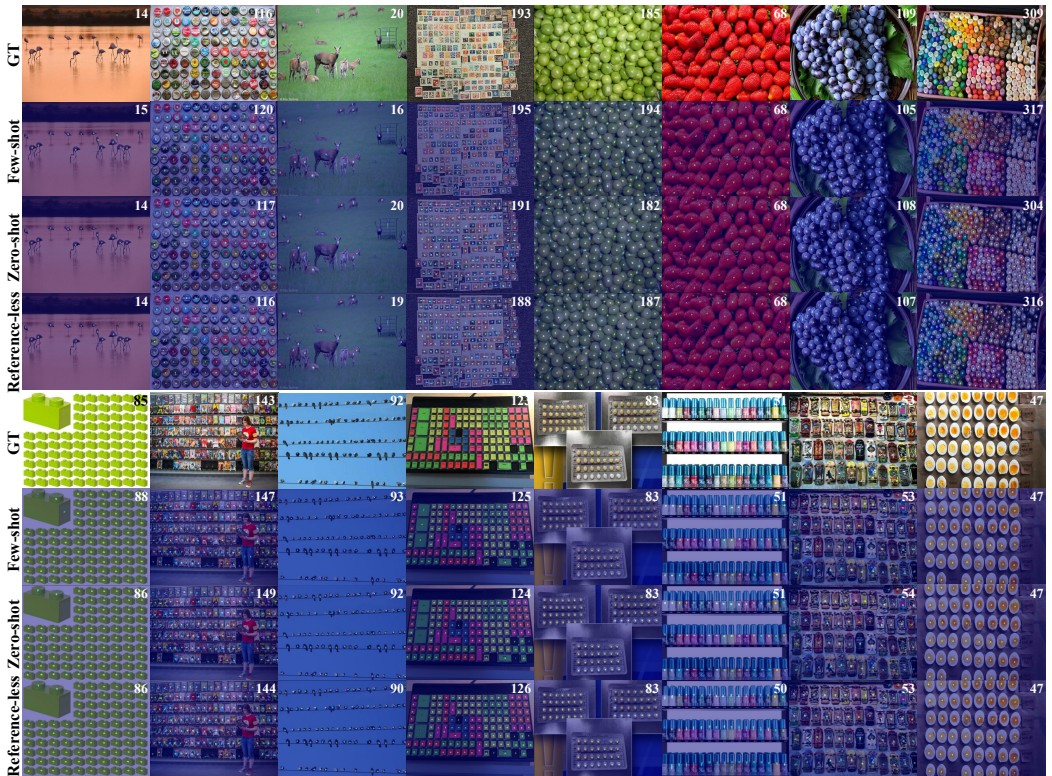

Figure 9: More qualitative results on the FSC-147 dataset.

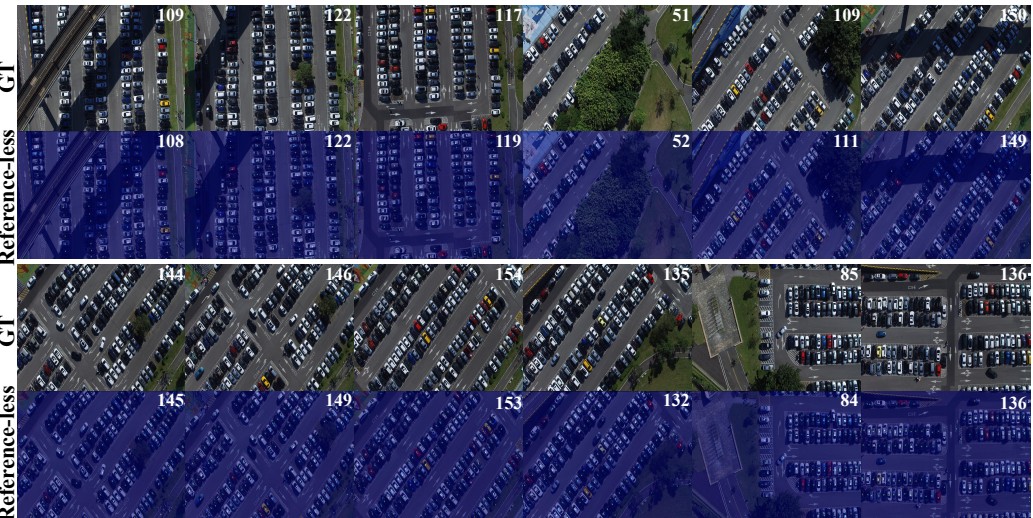

Figure 10: Qualitative results on the CARPK dataset.

