# OpenReview forum: "A Generic Class-agnostic Object Counting Network with Adaptive Offset Deformable Convolution"
_ICLR.cc/2025/Conference — ICLR 2025 Conference Withdrawn Submission_

### Official Review · Reviewer_w1ca · 2024-10-29

**Soundness:** 3
**Presentation:** 3
**Contribution:** 2
**Rating:** 6
**Confidence:** 4

**Summary:**

This paper designs a generic class-agnostic object counting network with Adaptive Offset Deformable Convolution (AODC), which focuses on the reference-less class-agnostic object counting task.

**Strengths:**

The outstanding performance of this paper is reflected in the experimental results.

**Weaknesses:**

1) The motivation is not clear, please explain why you focus more on reference-less tasks. Based on my understanding of this task, few-shot and zero-shot tasks based on text descriptions or visual exemplars are more practical. Users tend not to count how many objects there are in an image, but rather to count the number of specific categories. This is why recent research focuses more on these two types of methods. Based on this, please give application scenarios.

2) The presentation of Fig. 2 is confusing. It seems that the few-shot setting and the main branch are partially repeated. If they are different, what is the difference? Please reorganize this figure to make it easier for readers to understand.

3) Please explain why 4D convolution was used rather than 3D convolution.

4) It seems that deformable convolution has been used for counting tasks for a long time. Please explain the novelty of the proposed method and compare it with related methods, such as: ADCrowdNet: An Attention-Injective Deformable Convolutional Network for Crowd Understanding.

**Questions:**

NA

---

> ### Author Response · Authors · 2024-11-13
>
> Thank you for your valuable comments and suggestions. Due to the unsatisfactory results of the review, we have decided to withdraw our submission afterward. However, we still provide the following responses to your questions:
>
> 1. What you say makes sense, but it is more costly to perform additional labeling work comparisons, and I think it makes sense to find a method that does not require additional labeling operations. If in a scenario similar to amount control, All objects must not add up to exceed the limit, our method still has some application scenarios. More importantly, we want to use this work to bring people a way of thinking that there is no need for reference to recognize various information about an object. We believe that someday in the future there will be a way to recognize object categories and even more information without reference.
>
> 2. In fact, the zero-shot is exactly the same as the reference-less setting except for an additional feature fusion operation, so we did not make a detailed distinction in order to avoid duplication. However, we will take your suggestion into full consideration and make corrections in subsequent studies.
>
> 3. Because since the image is 2D, then the resulting self-similarity map is 4D, 3D convolution cannot be used in 4D maps.
>
> 4. Our contribution lies mainly in the use of adaptive offsets, and comparable convolution is just a technical tool to achieve convolution kernel expansion.

---

### Official Review · Reviewer_wuEu · 2024-10-30

**Soundness:** 2
**Presentation:** 2
**Contribution:** 2
**Rating:** 5
**Confidence:** 4

**Summary:**

This paper proposes a unified framework for the Object Counting problem, in which firstly proposed for reference-less Object Counting and can be easily adapted to the few-shot and zero-shot setting. Through the experiments, the paper successfully shows the method has good performance over each setting.

**Strengths:**

- This paper has a clear structure, motivation, and experiment to validate the effectiveness of the proposed methods.
- The idea of having an unified framework for few-shot, zero-shot and reference-less setting from the reference-less setting is new, but is not significant (see the weakness).

**Weaknesses:**

**Significance of the contribution**:
- The ideas of the generalized framework are good, but it is not new. LOCA [1] can perform object counting for referenceless and few-shot settings, as indicated in the paper. With appropriate modification, for example, injecting the cross-attention with CLIP, it may gain the capability of zero-shot counting.
- In the ablation study (Tab 3), there are 3 main factors contribute to the good performance of the model. 1) The Generalized Loss, which is already used in [2], 2 )the Deformable Convolution part, which is the main focus of the paper, is not clearly written. Therefore, it makes the contribution less significant.

**Clarity**: It is not well clearly written in the Deformable Convolution part. In Fig 2, there is a branch that connects from $O_{local}$ to the Deformable Conv, but is not mentioned in the Deformable Convolution Section, therefore, it is hard to understand the reason why the paper needs to predict $O_{local}$ and $O_{adaptive}$ seperately. In addition, there is a term "fusion convolution" in L 223-224, which is not clearly defined.

**Experiment**:
- The results for the Few-shot setting is not good as LOCA [1], even LOCA use only L2 loss for the training of the model, and in the paper, the ablation study (Table 3) shows that the Generalized Loss can achieve better performance compared to L2 loss.
- Compared to previous works, for example LOCA [1], the paper added a deformable convolution part. And to mitigate the deformable convolution, the paper tries to add some 4D Convolution to predict the offsetmap. I think the paper should report the time and GPUs memory running in the whole process, is it worth to run the big module, since the performance in the few-shot setting is not good compared to LOCA [1].

[1] Nikola Djukic, Alan Lukezic, Vitjan Zavrtanik, and Matej Kristan. A low-shot object counting network with iterative prototype adaptation. pp. 18872–18881, 2023.
[2] Wei Lin, Kunlin Yang, Xinzhu Ma, Junyu Gao, Lingbo Liu, Shinan Liu, Jun Hou, Shuai Yi, and Antoni Chan. Scale-prior deformable convolution for exemplar-guided class-agnostic counting. In Proceedings of the British Machine Vision Conference, 2022.

**Questions:**

1. What is $O_{local}$ and $O_{adaptive}$ represented for? And how do we incoporate only $O_{local}$ in the Deformable Convolution block?
2. Why do not use the cross attention in the few-shot setting similar to the zero-shot setting, since we can treat the features of the examplars as the CLIP features?

---

> ### Author Response · Authors · 2024-11-13
>
> Thank you for your valuable comments and suggestions. Due to the unsatisfactory results of the review, we have decided to withdraw our submission afterward. However, we still provide the following responses to your questions:
>
> The fact is that using GL loss on loca doesn't give much of a boost to the model because GL doesn't necessarily apply to all methods. The reason why it gives a huge boost to AODC is that the key points of our method fit very well with a point-labeling based loss like GL.
>
> The 4D convolution we use is in principle actually a combination of multiple 2D convolutions, which does not result in much computational cost. In the appendix we also give data on the computational cost of the whole model relative to other SOTA methods, proving that our method is not slow.
>
> $O_{local}$ is the offset computed directly from the feature itself, utilizing the scale information that the feature itself may contain, while $O_{offset}$ is computed based on the self-similarity, reflecting the category-agnostic offset computation. If only $O_{local}$ is available, it is likely that the model will not be able to recognize well categories that have not been seen during training.
>
> Cross-attention is meant to incorporate text information, and there is no additional text information in the few-shot setting, so cross-attention cannot be added or it would be unfair.

---

### Official Review · Reviewer_qByJ · 2024-11-03

**Soundness:** 2
**Presentation:** 2
**Contribution:** 3
**Rating:** 5
**Confidence:** 4

**Summary:**

The paper introduces a novel approach for class-agnostic object counting, focusing on counting unseen object classes using an Adaptive Offset Deformable Convolution (AODC) network. The key innovation lies in generating self-similarity maps from image features, followed by a 4D convolution to create adaptive offsets for deformable convolution. This design enables the model to capture multi-scale objects effectively. The method is adapted for zero-shot and few-shot counting and achieves state-of-the-art results across various datasets and counting scenarios.

**Strengths:**

1. The proposed AODC framework, grounded in adaptive offset deformable convolution, presents a novel approach for capturing spatial structures of objects at different scales. It achieves impressive results across reference-less, zero-shot, and few-shot settings, showing its adaptability.
2. Extensive experiments across multiple datasets, including FSC-147, CARPK, and a COCO subset, underscore AODC’s effectiveness and generalization capability.

**Weaknesses:**

1. This paper focuses on the design of deformable convolution; relevant work should be discussed in the literature review.
2. The architecture in Fig. 2 lacks clarity and explanation. For example, $F$ is related to multiple components, but the arrows lack further clarification.

**Questions:**

1. The ablation study only shows the reference-less setting. Could additional analysis be provided to show the impact of different components across other settings?
2. Could you provide a visualization comparing the effects of using vs. not using Adapt-O, similar to the format used in Fig. 6?
3. In Tab. 2, the few-shot settings on Val-COCO and Test-COCO seem to have moderate performance. Is there any explanation?
4. From my observation, most images in the test dataset are dominated by a single object category (e.g., an image filled with strawberries).
The proposed method, which utilizes a self-attention like mechanism, may lead the model to focus on all objects within an image, potentially achieving high performance on such datasets without truly understanding the similarity between exemplars and the image.
Could you demonstrate the model's ability to distinguish different object categories within the same image? (e.g., using exemplars of different categories, such as apples and strawberries, on an image containing various fruits.)

---

> ### Author Response · Authors · 2024-11-13
>
> Thank you for your valuable comments and suggestions. Due to the unsatisfactory results of the review, we have decided to withdraw our submission afterward. However, we still provide the following responses to your questions:
>
> Given our impending withdrawal, please forgive us for not being able to provide you with additional experimental data. The answer to your question about recognizing a single category in a sample of multi-category objects is yes, because specific exemplar images are provided, and the category information obtained from the images is very different from that of the other categories.

---

### Official Review · Reviewer_8Uv1 · 2024-11-03

**Soundness:** 2
**Presentation:** 2
**Contribution:** 2
**Rating:** 3
**Confidence:** 3

**Summary:**

In this paper, the authors propose Adaptive Offset Deformable Convolution (AODC), which aims to recognize objects of different scales in a same sample. It can be also extended to zero-shot and few-shot settings, achieving state-of-the art performance across different counting datasets.

**Strengths:**

1. Handling objects of varying scales is a critical challenge in counting.
2. The method achieves state-of-the-art performance on different counting datasets.

**Weaknesses:**

1. The motivation of the paper is not particular clear. Although the introduction mentions that the objects for counting are of varying scales, and the proposed method can recognize these variations, there is not enough evidence to show the effectiveness across objects of different sizes. Most of objects from FSC-147 dataset are similar in size, and the objects shown in Fig. 4 and Fig. 8 are also of similar sizes. These results may not sufficiently validate the method's ability to effectively handle diverse scales.
2. The novelty is limited. The use of deformable convolution for object counting has been previously proposed in [1]. The only difference from [1] is the replacement of a fixed scale with multiple scales, which is not a significant contribution.
3. The performance of models in the few-shot on COCO appears to be inferior to that of comparison methods, which raises concerns about the effectiveness in this setting.

[1] Lin et al. Scale-prior deformable convolution for exemplar-guided class-agnostic counting. BMVC 2022

**Questions:**

Can the authors show the results of the images in Fig. 5 under zero-shot setting?

---

> ### Author Response · Authors · 2024-11-13
>
> You reviewing my submission again? Same scores, same comments? Did you get rejected too? That's great!
>
> I don't have anything else to say, except that I wish you all the same reviewers you've encountered with your future submissions as yourself.

---

### Official Review · Reviewer_NkPb · 2024-11-03

**Soundness:** 2
**Presentation:** 2
**Contribution:** 1
**Rating:** 5
**Confidence:** 4

**Summary:**

This paper tackles the problem of class-agnostic counting. For reference-less setting, the proposed method computes self-similarity map to obtain shape and size information of the objects, and applies 4D convolution to learn adaptive offsets for deformable convolution. For few-shot and zero-shot settings, the proposed method uses cross-similarity maps instead of self-similarity maps. The proposed method is evaluated on the three settings of class-agnostic counting.

**Strengths:**

1. The proposed method is generalizable to different settings in class-agnostic counting.
2. The proposed framework shows good performances on different settings.

**Weaknesses:**

1. My main concerns are with the technical contributions and the novelty. Though it can be important to adaptively learn object shapes and structures, the proposed method seems to learn adaptive offsets for convolution kernels in a quite straightforward way. It seems like a composition of existing techniques (e.g., deformable convolution, 4D convolution, generalized loss, etc.).
2. The writing of the method section is not very clear. I can understand the high-level pipeline, but I'm confused about the details, especially the deformable convolution module.
3. There lacks some analysis on why using adaptive offsets and deformable convolution improves the performance, e.g., why the authors chose to adopt deformable convolution instead of other adaptive modeling techniques?
4. The ablation study shows that the adopted generalization loss improves the performance a lot, but it is not proposed by this paper and is not claimed as a contribution of this paper. It is also not adopted by some of the compared methods.

**Questions:**

1. Would the performance of the proposed framework be affected by the background in the input image? As mentioned in Line 54-55, the target objects become "more obvious when the object is in a background with a large difference in colour compared to itself".
2. Can the proposed method deal with objects with various sizes? In class-agnostic counting, a major challenge is the varying sizes and colors of the target objects in the same image. I'm curious about whether the adaptive offsets learned can adjust to different sizes.

---

> ### Author Response · Authors · 2024-11-13
>
> Thank you for your valuable comments and suggestions. Due to the unsatisfactory results of the review, we have decided to withdraw our submission afterward. However, we still provide the following responses to your questions:
>
> 1. In fact, not only for our method, but for any method, the performance is worse on those samples where the background and foreground colors are similar. This is an intuitive conclusion, since it is really hard to go about distinguishing the objects that need to be recognized in a scene with similar colors.
>
> 2. Honestly, for those samples where the difference in object scales is not particularly large, our method is able to recognize the scales of individual objects very well, as we show in Fig.6 and Fig.8. However, for some samples with extreme scale differences may not be recognized very well, because the convolution itself is localized and the calculated offsets cannot be too large or too small.

---

### Official Review · Reviewer_ohYm · 2024-11-04

**Soundness:** 2
**Presentation:** 3
**Contribution:** 2
**Rating:** 5
**Confidence:** 4

**Summary:**

The paper proposes a generalized counting framework with the proposed adaptive offset deformable convolution (AODC). By augmenting pixel-wise self-similarity map as offset to deformable convolution, it effectively enhances localization that converges to the center of each object. In particular, the use of 4D convolutions transforms self-similarity map into adaptive offset. Along with attention mechanisms for feature refinement and generalized loss(GL) to supervise dot labels, the proposed, unified framework acquires SOTA across few-shot, zero-shot, and reference-less settings.

**Strengths:**

1. The empirical result is impressive, especially on zero-shot and reference-less settings. Experiments are comprehensive. It shows the generalizability of the framework.
2. The use of 4D convolutions to exploit the local information of self-similarity map to capture object-level information and transform into offsets is insightful, especially under reference-less setting.
3. The paper is relatively easy to follow

**Weaknesses:**

1. The novelty is somewhat limited. The use of deformable convolution has been used in SPDCN. The author’s proposed contribution is to improve the offset to be adaptive on self-similarity maps. Attention for feature refinement is used in existing counting works such as LOCA. While GL is not a proposed contribution by the authors, it not new as well
2. My main concern lies in the impact of GL on the improved performance. Shown in the ablation studies (Table 3), the reference-less performance without GL is worse than prior works that uses MSE loss, which suggest that GL is the core factor for improvement. The performance achieves SOTA after employing GL, which is not a fair comparison given other models are trained on MSE, a suboptimal solution of GL. It would be fairer if you compare against prior works trained on GL. Please provide explanations on your setup
3. The performance on CARPK is not SOTA under few-shot setting. CounTR yields better performance than your proposed method (MAE: 5.75, RMSE: 7.45)
4. Missing newer counting methods such as [1] (from ECCV 24’)

[1] Jiang, Q., Li, F., Zeng, Z., Ren, T., Liu, S., & Zhang, L. (2024). T-rex2: Towards generic object detection via text-visual prompt synergy. In European Conference on Computer Vision (pp. 38-57). Springer, Cham.

**Questions:**

1. Regarding the impact of GL loss on few-shot setting, it would be beneficial to provide the ablation study to verify whether the SOTA result is benefitted by GL or the proposed AODC framework?
2. Could you visualize the ablation study to concretely demonstrate the role of each component?
3. (Minor comment) Is the method scalable to ViT-based backbones, in which the ability to capture global dependencies could benefit in identifying counting targets?
4. (Minor Comments) Table 4 discusses the choice on the number of attention blocks and the phenomenon of overfitting. While stacking too many attention blocks worsens the validation performance, it is only overfitting if the training loss is continually decreasing. I am not certain whether this is the case, and if not, I would suggest remove such wording. I am more inclined to state that excessive number of attention blocks strengthens dominant features and yet filter out less dominant features that might correspond to target objects.

I am willing to raise my score if the above questions (especially GL) are adequately addressed.

---

> ### Author Response · Authors · 2024-11-13
>
> Thank you for your valuable comments and suggestions. Your suggestions are indeed very informative, however, unfortunately, due to the unsatisfactory results of the review, we have decided to withdraw our submission afterward. However, we still provide the following responses to your questions:
>
> 1. First of all, regarding the GL loss, which is the most important issue for you, GL does have advantages over MSE in some aspects, however, not all models are suitable for GL. We have done similar experiments on some models before, and found that most of the methods do not have obvious improvement after using GL, and even some models have decreased in performance. The reason why GL can have a very obvious effect on our method is that the key part of our method, i.e., offsetting the convolution kernel at the center of the object to cover the whole object, fits well with the scenario of GL, which is based on the point labeling loss function.
>
> 3. Because ViT will cut the image into large chunks, which will greatly lose the spatial structure of the image, which is the key of our method, serialization models like ViT are not very suitable for our method.
>
> 4. Yes, during the training process, the loss on the training set is still decreasing, but the performance on the validation set gets worse, so we tend to use "overfitting". But your idea is also very reasonable, we may consider your suggestion in the follow-up research, thanks.

---

### Note · Authors · 2024-11-26

I have read and agree with the venue's withdrawal policy on behalf of myself and my co-authors.